

# Intelligent decision making for energy efficient fog nodes selection and smart switching in the IOT: a machine learning approach

Rahat Ullah[1], Muhammad Yahya[2], Leonardo Mostarda[3], Abdullah Alshammari[4], Ahmed I. Alutaibi[5], Nadeem Sarwar[6], Farhan Ullah[7] and Sibghat Ullah[8]

[1] Institute of Optics and Electronics, Nanjing University of Information Science and Technology, Nanjing, China
[2] Qurtaba University, Peshawar, Pakistan
[3] Computer Science School of science and technology, University of Camerino, Camerino, Italy
[4] College of Computer Science and Engineering, University of Hafr Albatin, Hafr Albatin, Saudi Arabia
[5] Department of Computer Engineering, Majmaah University, Majmaah, Saudi Arabia
[6] Department of Computer Science, Bahria University Lahore Campus, Lahore, Pakistan
[7] School of Software, Northwestern Polytechnical University, Xian, China
[8] National Research Center for Optical Sensors/Communications Integrated Networks, School of Electronic Science and Engineering, Southeast University, Nanjing, China

Corresponding authors
Muhammad Yahya,
yahya.cs07@gmail.com
Sibghat Ullah, sibghat@bupt.edu.cn

## ABSTRACT

With the emergence of Internet of Things (IoT) technology, a huge amount of data is generated, which is costly to transfer to the cloud data centers in terms of security, bandwidth, and latency. Fog computing is an efficient paradigm for locally processing and manipulating IoT-generated data. It is difficult to configure the fog nodes to provide all of the services required by the end devices because of the static configuration, poor processing, and storage capacities. To enhance fog nodes' capabilities, it is essential to reconfigure them to accommodate a broader range and variety of hosted services. In this study, we focus on the placement of fog services and their dynamic reconfiguration in response to the end-device requests. Due to its growing successes and popularity in the IoT era, the Decision Tree (DT) machine learning model is implemented to predict the occurrence of requests and events in advance. The DT model enables the fog nodes to predict requests for a specific service in advance and reconfigure the fog node accordingly. The performance of the proposed model is evaluated in terms of high throughput, minimized energy consumption, and dynamic fog node smart switching. The simulation results demonstrate a notable increase in the fog node hit ratios, scaling up to 99% for the majority of services concurrently with a substantial reduction in miss ratios. Furthermore, the energy consumption is greatly reduced by over 50% as compared to a static node.

# INTRODUCTION

Wireless sensor networks (WSNs) are commonly used in the Industrial Internet of Things (IIoT) to enhance productivity and efficiency in manufacturing. Sensor network virtualization and overlay networks are particularly interesting in IIoT (*Li & Savkin, 2018*; *Bughin, Chui & Manyika, 2010*). The WSNs and communication technologies like Bluetooth and ZigBee have improved fault prediction, quality control, and localization in industrial settings. Data from sensors, machines, and actuators can be collected and accessed through the IIoT (*Rahman et al., 2019*; *Stergiou et al., 2018*). The IoT is a network of physical objects with microchip technology, software, sensors, and network connections that can exchange data. The IoT will have a significant impact on consumers' daily lives and activities (*Choi & Ahn, 2019*; *Kumar, Dubey & Pandey, 2021*; *Yousefpour et al., 2019*).

With the emergence of IoT, the number of authorized end-users, network fog, and access devices has increased, such as smart cities, smart industries, smart grid stations, smart meters, smart police stations, and smart traffic control systems (*Ahmad & Ranise, 2019*). The IIoT increases economic values and efficiency and reduces human intervention. Traditional IIoT applications use a centralized approach (typically a cloud) for extensive data processing and storage. However, with the tremendous growth of industrial data, sending all of this data to the cloud on congested networks is becoming increasingly difficult (*Trequattrini et al., 2016*). Furthermore, due to the ultra-low latency requirement, monitoring and managing crucial and event-triggering equipment makes it impossible to store and process data in the cloud (*Yi et al., 2015*; *Clohessy, Acton & Morgan, 2014*). The use of cloud and IoT is growing rapidly, affecting various industries like smart cities, smart power stations, smart health care, and smart manufacturing. The cloud's centralized nature does not satisfy the decentralized nature of IoT's requirements (*Yu et al., 2017*; *Li, Xu & Zhao, 2015*).

Fog computing is a network of devices that provide computing services to end-users in their local area, reducing bandwidth and energy consumption by processing data in the network fog instead of transmitting it to the centralized cloud (*Condry & Nelson, 2016*; *Ping et al., 2017*; *Vinod et al., 2018*). It is considered the third revolution in IT and can bridge the gap between the physical and digital worlds, enhancing lifestyle and industry productivity. Smart city architecture with fog nodes can improve worker safety and protect the working environment from potential risks of intelligent transportation and smart grid stations (*Alhmiedat, Taleb & Bsoul, 2012*; *Maier, Ebrahimzadeh & Chowdhury, 2018*; *Shu, Mukherjee & Wu, 2016*). The rapid growth of industry data makes it challenging to move all manufacturing data to the congested network, and ultra-low latency devices require monitoring and management that cloud processing cannot provide (*Omarov & Altayeva, 2018*; *Cabra et al., 2017*). The integration of fog computing with IoT is needed to enable ultra-low or predetermined latency for time-sensitive applications (*Grabia et al., 2017*). Clouds are shifting towards smaller, more integrated communication and computing closer to users, sensors, and actuators. Cloud of Things applications do not support real-time or latency-sensitive applications due to high latency on the cloud, negatively affecting Quality of Service (*Khalil et al., 2014*; *Xu et al., 2016*; *Alhmiedat, 2015*).

This research article presents an effective and efficient fog node reconfiguration model to address the challenges mentioned above. This study aims to overcome the challenges related to data latency and energy consumption in fog computing environments. Our approach focuses on bringing fog services in close proximity to consumer devices and sensors, enabling them to participate actively in data processing with minimal latency. By utilizing the computing capabilities of these devices, we significantly reduce the amount of data that needs to be transmitted to the cloud. This reduction in data transmission leads to a notable decrease in communication latencies, as the distance traveled by data packets is minimized. By leveraging the computing power of nearby consumer devices and sensors, we can process data locally, eliminating the need for round trips to the cloud for every processing task. This results in faster response time and improved system performance.

Moreover, our approach contributes to energy efficiency by reducing the number of active fog nodes. Traditional fog computing architectures typically employ many fog nodes distributed across the network, consuming substantial energy. By leveraging nearby consumer devices and sensors, we can consolidate processing tasks and minimize the required active fog nodes. This reduction in the number of fog nodes translates to lower energy consumption, contributing to sustainability efforts and reducing operational costs.

The rest of the article is organized as follows: "Introduction" contains the introduction and literature review. "System Model" discusses the proposed methodology. The third section describes our proposed model and discusses the achieved results. In the end, the article is concluded with a conclusion in "Conclusions".

## SYSTEM MODEL

The decision tree (DT) algorithm for the fog node generates decisions based on its experience to minimize energy costs in a dynamically changing environment. The algorithm is trained on a provided dataset to enable intelligent decision-making in response to sensor node requests. DTs can evaluate all possible outcomes and handle both continuous and categorical variables effectively. Additionally, they provide clear indications regarding the relative importance of fields for prediction or classification purposes. The algorithm categorizes input data for classification and makes decisions based on the classified data for services. The output layer displays the desired decision, as shown in Fig. 1.

The algorithm provides the best decision with a particular event's maximum hit and minimized miss ratio. The hit ratio refers to the correctly predicted value of a required service, whereas the miss ratio refers to the missed value or inaccurately predicted value. Furthermore, the proposed algorithm has achieved comparatively low latency with minimum energy consumption. The architecture of the proposed model is presented in Fig. 2, which illustrates the design of smart cities with fog nodes using a machine learning algorithm. Health care, smart transportation, smart water systems, smart industry, and smart grid stations sense the environment in the forest; each system has its fog node, as can be seen in Fig. 3. Fog computing is a geographically distributed paradigm that extends cloud computing and networking capabilities to the network edge, bringing computing

**Input Layer**   **Hidden Layer**   **Output Layer**

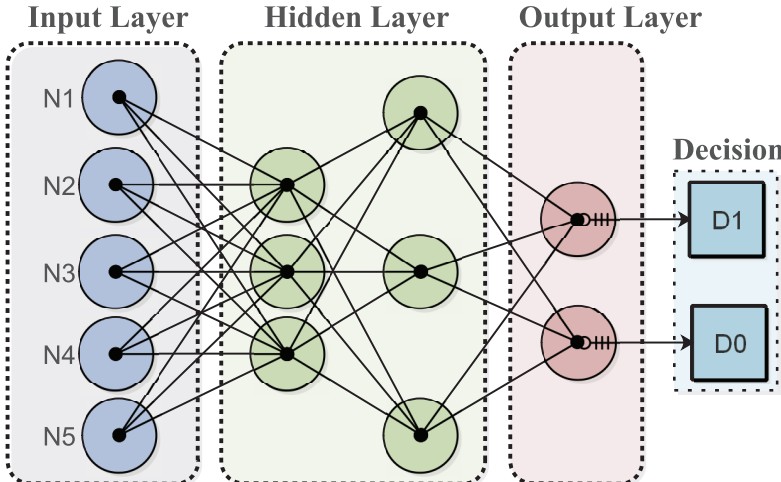

**Figure 1 Multilayer architecture of the proposed model.**

and networking capacity closer to end-users and IoT devices. The proposed algorithm is shown in Table 1.

The algorithm initiates by setting up initial parameters and variables. $E_n^t$ represents the hosting decision of fog nodes at time t; this variable stores the current fog node hosting decisions. $E_n$ is the set of fog nodes, denoted as $n = \{1,\ldots, N\}$, encompassing all available fog nodes. $D_t$ represents the decision tree used for event prediction which is a machine learning algorithm that helps in predicting events based on certain attributes. SN represents the set of sensor nodes. $Y_x \epsilon R^n$ i = 1 represents the training vector and training data used to build and train the decision tree. The static energy consumption of fog nodes is represented by $E_n^0$, whereas, the unit energy consumption for one cycle in fog node n is represented by $K_n$. The Endorse step in the algorithm indicates that fog node reconfiguration and efficient deployment are carried out. If e = 1, the algorithm will check if the event (prediction) has occurred and then initialize the event prediction using $D_t$ which initializes the event prediction process using the decision tree. N = {ozone, particulate matter, carbon monoxide *etc.*} specifies the attributes/features used for event prediction. In this case, the attributes are related to environmental factors like ozone, particulate matter, carbon monoxide, *etc.* In the case of an event occurrence, the sensed values of the attributes are compared with the predefined threshold values to determine if an event has occurred. Afterwards, the entropy of the event prediction is calculated which measures the uncertainty or randomness of the predicted events. The energy consumption of fog nodes based on the current fog node hosting decisions, the static energy consumption, and the energy consumed by services requested by fog nodes is calculated using *Entropy* $\sum_{i=1} - p * ln2(pi)$ . Furthermore, the energy consumption of fog nodes is computed based on the services requested by them. On the contrary, if no event (prediction) has occurred, the value of e will be 0, and the else statement will be executed, the decision will be discarded, and appropraite actions are taken even for null events as well.
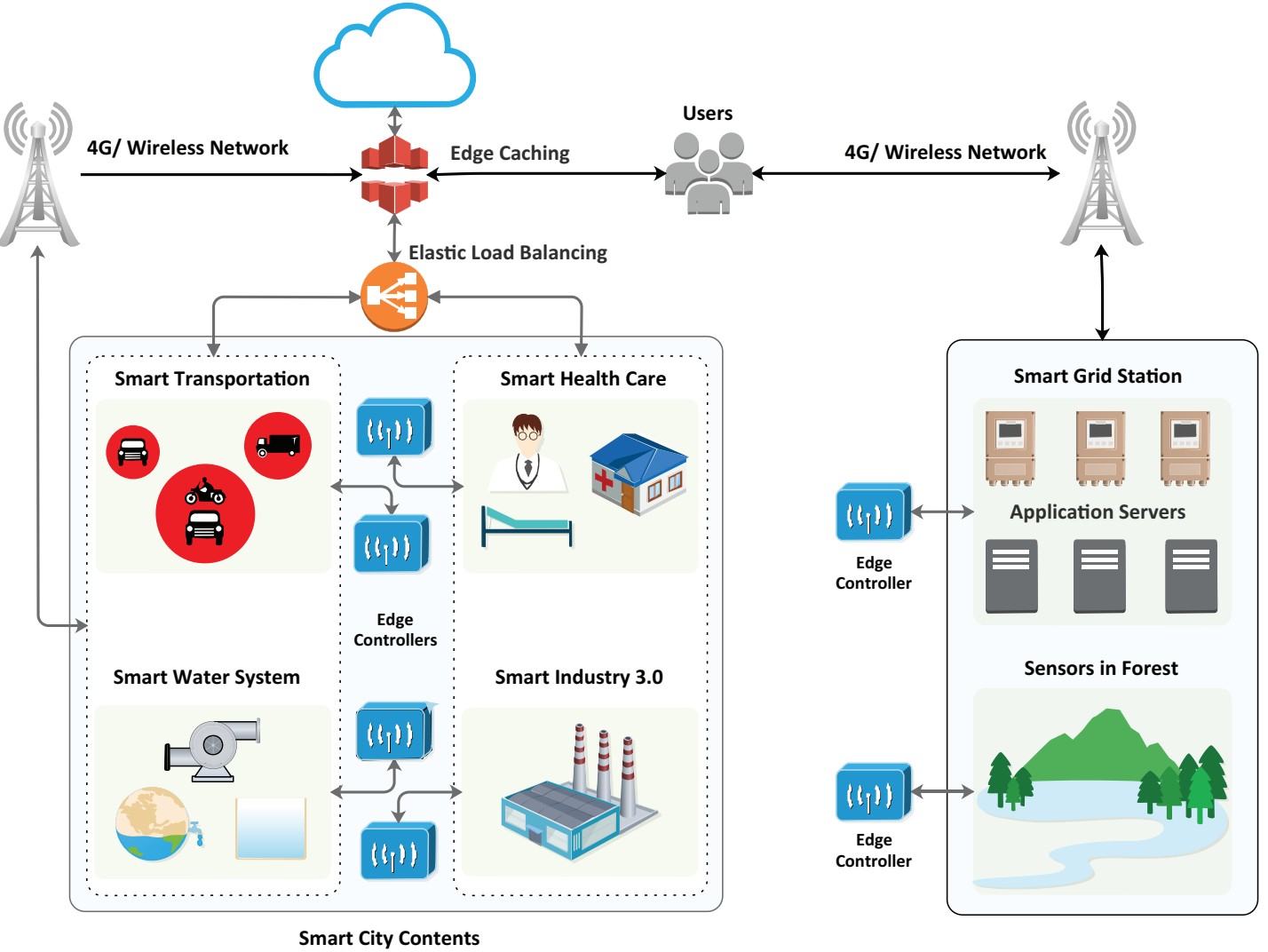

**Figure 2 Proposed architecture of the smart city with fog nodes.**

Overall, the algorithm utilizes a decision tree for event prediction and makes decisions on fog node selection and switching based on the occurrence or non-occurrence of events and the energy consumption considerations.

The collection of maximum and minimum values of a particular service from the whole dataset is calculated using the given equation.

$$\forall \text{ Data set} = \frac{Th + 1}{2} \tag{1}$$

where *Th* shows the threshold value of a particular service.

The threshold range values are identified, and the analysis of sensor threshold values is conducted to determine whether they indicate a normal state or have reached hazardous levels. This determination is made using Eqs. (2) and (3). The sensing values are

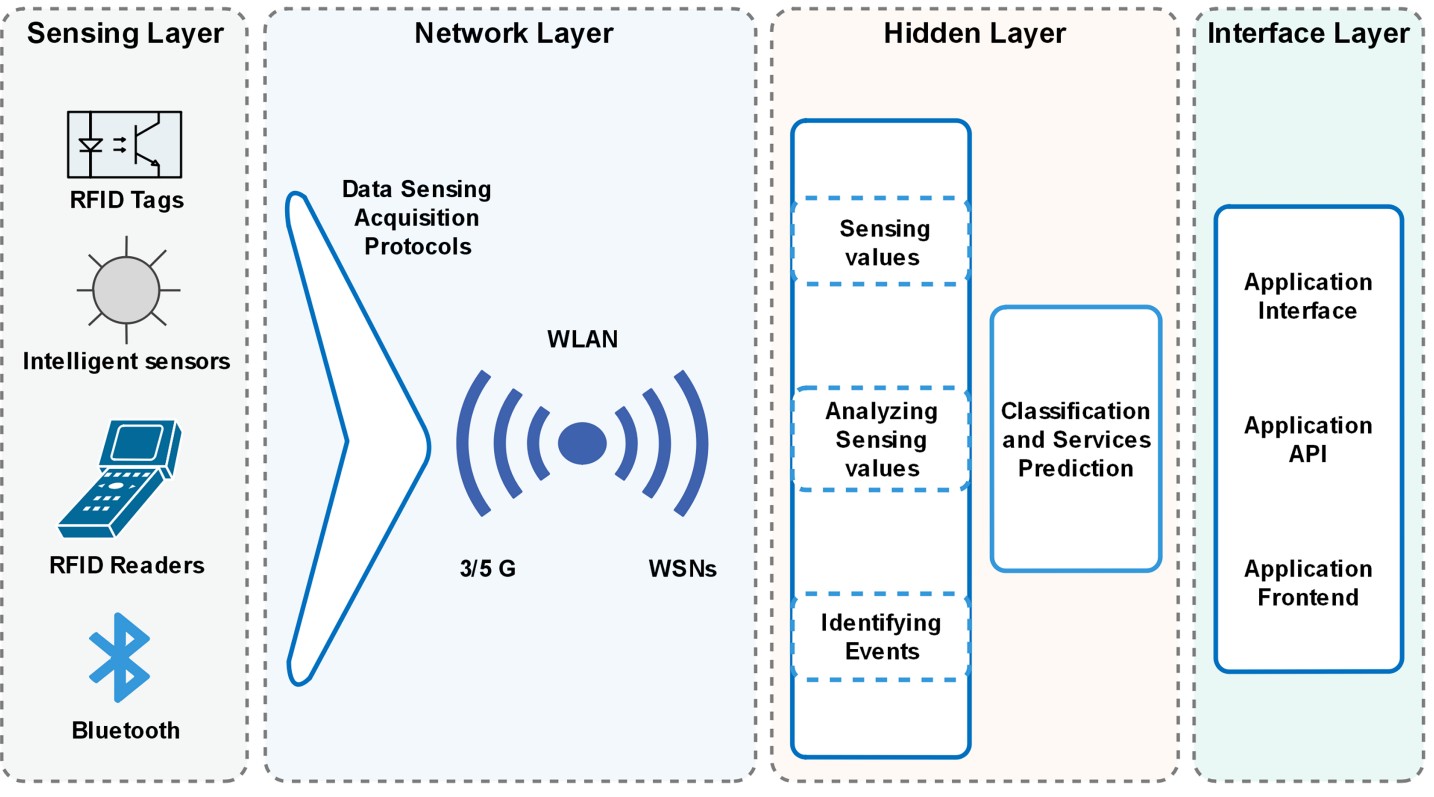

**Figure 3 Architecture of the proposed model.**

categorized into two parts: either a hazardous state represented by the variable xi or a normal state represented by the variable x in Eq. (2). In Eq. (2), mi represents the proportion of the sensing values and takes the values of xi and x for testing purposes.

$$\sum_{i=1-}^{tn} \frac{mi}{n} \tag{2}$$

Equation (2) calculates the sum of ratios (mi/n) across a range of values from $i = 1$ to $tn$. Each term in the summation represents the ratio of a specific quantity ($mi$) to the total count ($n$) associated with that value.

$$\forall \sum_{i=1}^{n} \left( \frac{(xi - x)^2}{n} \right) \tag{3}$$

Equation (3) represents the calculation of the mean squared deviation of a set of values from their mean. The symbol $\forall$ denotes 'for all', and signifies that the following expression applies to all values or elements in the set. The starting index of the summation, denoted by $i$, is set to 1, indicating that the summation begins with $i = 1$. The variable $n$ represents the total number of values or elements in the set being measured. The expression $(xi–x)^2$ signifies the squared difference between each value ($xi$) in the set and the mean value $x$. The

**Table 1  Proposed algorithm.**

| Algorithm: Efficient fog node selection and smart switching based on DT algorithm. |
|---|

1:  Initialization:
2:  $E_n^t$  Fog nodes hosting decision at time t
3:  $E_n$  Set of fog nodes  n= {1, ..., N}
4:  $D_t$  Decision tree
5:  $E_n$  Number of fog nodes
6:  SN  Set of sensor node
7:  Hd  Hosting decisions
8:  $Y_x \epsilon$  Training vector
9:  $R^n$  Training vector
10:  i=1  Training vector
11:  $E_n^0$  Static energy consumption
12:  $K_n\, Cf_n^2$ (unit energy consumption for one cycle)
13:  Endorse:
14:      Fog node ($E_n$) reconfiguration
15:      Efficient fog node deployment
16:  If e = 1
17:      Decision step do
18:      Initialize event prediction using $D_t$ (p+ n−)
19:  Check attributes
20:      N= {Ozone, Particulate matter, Carbon monoxide *etc.…*}
           Where n = 10;
21:  Check threshold (TH) values of attributes (sensing)
22:      If sensing = Dataset [TH]
23:          The responding event occurs
24:      Else if sensing != Dataset [TH]
25:          e = 0
26:      Collect the whole fog node hosting decisions
27:      Calculate entropy for the whole prediction:
28:      *Entropy* $\sum_{i=1}\;\; -p * ln2(pi)$
29:      Hd = MnCm
30:      $E_n^t(a_t, b_t) = E_n^0 + K_n b_{t_n} \sum_K \mu k \lambda_{n,k}^t (a^t)$
31:      For ∀ E, calculate energy consumption occupied by fog nodes for services request
32:  Else if e=0
33:      Decision step discard
34:      No action is taken for null events

training vectors for the services of sensing environment are denoted as $Y_x \epsilon R^n$, activating the sensors (*e=1*) & label vector Z $\epsilon R^l$. The training variables are shown in Table 1, and samples with similar labels or goal values are grouped. The node variables for services are denoted as $E_n$ & $X_n$.

$$E_n^{left}(\theta)\; E_n^{right}(\theta) \tag{4}$$

Equation (4) describes a cost or error related to the left portion of the data following a split. The cost is calculated on a dataset of size 'n', as indicated by the subscript 'n'.

$$E_n^{left}(\theta)[(x, \ y) \mid xi <= tn ] \tag{5}$$

where $E_n^{left}(\theta)$ indicates the cost or error associated with the left subset of the data after a split. The subscript 'n' denotes that the cost is calculated on a dataset of size 'n,' and 'θ' represents the parameters or conditions used for the split. Similarly, the process described in Eq. (5) is executed for the right subtree splitting, as explained in the equation.

$$E_n^{right}(\theta) \ [(x, \ y) \mid xi <= tn] \tag{6}$$

The data is sorted at each node of the tree, and the splitting of node (n) to choose the optimum splitting attribute is discussed in Eq. (6). To assess the splitting attribute, it applies the gain ratio impurity approach, as given in Eq. (7):

$$C\left(E_n\theta\right) = \ \frac{E_n^{left}}{X_n}R\left(E_n^{left}(\theta)\right) + \ X_n^{right}/X_n R \ E_n^{right}(\theta)) \tag{7}$$

Subsets are created by recursively applying $E_n^{left}(\theta^*)$ & $E_n^{right}(\theta^*)$ until the predicted value permissible is reached $Xn$.

Targeting classification values and taking values 0,1….k-1 for each node can expressed as Eq. (8)

$$Pmk \ = \frac{1}{nm}\sum y \in Qm \ I\left(y \ = \ k\right) \tag{8}$$

where Pmk is the possibility that an example in a specific node $m$ of the decision tree belongs to class $k$. The inverse of the number of examples in the subset $Qm$ that reach node $m$ is expressed as $\frac{1}{nm}$. This stabilizes the sum of indicator functions to calculate the probability. $\sum y \in Qm$ denotes the summation of all examples y that are part of the subset $Qm$. $I (y = k)$ serves as an indication that evaluates to (1) if the example $y$ falls under class $(k)$ and to (0) if it does not. If node $n$ is a terminal node, the proportion of class $k$ observations in that node is set to $Pmk$ to predict targeted values for this region.

$$Entropy \sum_{i=1} \ -p * ln2(pi) \tag{9}$$

Here, pi indicates the occurrence of particular services, introducing manipulative ambiguity in the predicted data. The accuracy of the predicted and sensed data is calculated using Eq. (9).

The attributes mentioned in Eq. (10) have been used in our research to calculate the uncertainty of the datasets by determining the odds of various outcomes. A decision tree can be implemented to classify records with uncertain values. To calculate these uncertain values, the entropy (Eqs. (10)–(12)) compute the impurity of each service.

Values (Services) = Ozone, Particulate_matter, Carbon_monoxide, Sulfur_dioxide, Nitrogen_dioxide

$$S = [p+, \; n-] \; \text{Entropy}(s) = -\frac{p}{10}\frac{\ln p}{10} - \frac{n}{10}\ln\frac{n}{10} \tag{10}$$

$$S_{Values(Services)} \; [p+, n-] \; \text{Entropy} \; (S \; x \ldots \ldots .n) \; = -p1\frac{p1}{10} \ln \frac{p1}{2} \; \frac{ln}{2} \ln \frac{n1}{2} \tag{11}$$

$$\text{Gain (Services)} = \text{Entropy}(s)\text{-}\underline{\quad} \sum \underline{\quad}\frac{SV}{S} \; \text{entropy}(sv)$$

$$\vee \in (ozone, P \; matter, \; C \; mono, \; S \; dio, \; N \; dio) \tag{12}$$

The symbol S in Eq. 10 refers to the entropy of the entire dataset or subset with the symbols. Entropy is a measure of impurity or uncertainty in a set of data. The weighted sum of probabilities for each class in the collection is calculated using the entropy formula, where 'p+' stands for the proportion of positive examples and 'n-' for the proportion of negative examples.

$$\text{Gain}(S - \text{Services}) = \text{Entropy} \; (s) - \frac{p2}{t10 \; entropy \; (S \; ozone)}$$
$$- \frac{-1}{t10 \; entropy \; (S \; P \; matter)}\frac{1}{t10 \; entropy \; (S \; C \; mono)} \tag{13}$$
$$\times \frac{1}{t10 \; entropy \; (S \; S \; dio)}\frac{-1}{t10 \; entropy \; (S \; N \; dio)}\frac{-1}{t10}$$

The gain (Services) in Eq. (13) represents the information gain for a specific attribute or subset, denoted as Services. Information gain measures how much the knowledge of a particular attribute or subset reduces the uncertainty in the overall dataset. The formula given as Entropy(s) – $\Sigma$(SV/S) * Entropy (sv) calculates the gain. $SV$ represents the size of the subset or attribute Services, while $S$ represents the size of the overall dataset. Entropy (sv) represents the entropy of each subset or attribute (sv) within the category of services. The summation is taken over all possible attributes or subsets within Services, such as ozone, particulate matter, carbon monoxide, sulphur dioxide, and nitrogen dioxide.

## Energy consumption

As long as fog node n is turned on, $E_n^o$ n is the static energy consumption independent of workload. $\kappa n = cf^2{}_n$ is the unit energy consumption for a single CPU cycle, determined by the CPU architectural parameter $c$ and the CPU frequency $fn$. The term $b_n^t \sum_k \mu k \lambda^t{}_{n,k}$ $(a^t)$ denotes the total number of CPU cycles needed to handle the service requirement of fog node $n$.

$M_n$ CM denote set of Sensors Nodes (SNs)
Set of SNs $M = [1, , , , M]$ $\qquad\qquad$ (14)
For Fog Nodes we denote it with $N = [1, , , , N]$

In our research, we compute the energy consumption using an intuitive and widely used model called the First Order Radio Model (FORM). Table 2 lists the FORM parameters used in conducting our research and the notation of simulation parameters are listed in

**Table 2 First-order radio model for energy consumption analysis parameters.**

| | |
|---|---|
| $E_{elec}$ | Electronics that transmit and receive data using the same amount of energy |
| $\varepsilon amp$ | Consistent consumption of energy for transmitter |
| $k$ | Number of Bits in a single packet |
| $d$ | Overall transmission distance |
| $r$ | Area of communication |
| $E_{tx}$ | The amount of energy used to transmit a k-bit packet |
| $E_{RX}$ | The entire energy used to receive a packet of k bits |
| $E_{ij}$ | Consumed energy to communicate k bits |

**Table 3 Notation description of the simulation parameters.**

| Notation | Description |
|---|---|
| $E_n^t$ | Fog nodes hosting decision at time t |
| $E_n$ | Set of fog nodes $n = \{1, ..., N\}$ |
| $D_t$ | Decision tree |
| $Y_x \in$ | Training vector |
| $R^n$ | Training vector |
| I=1 | Training vector |
| $E_n$ | Number of fog nodes |
| SN | Set of sensor node |
| Hd | Hosting decisions |
| $E_n^0$ | Static energy consumption |
| $K_n$ | $Cf_n^2$ (unit energy consumption for one cycle) |
| Networks area | 500 m * 500 m |
| Fog nodes | 10 |
| Sensors nodes | 200 |
| Number services | 5 |

Table 3. As a result, the energy consumption for delivering a k-bit message to a distant receiver using FORM can be expressed as:

$$E_{tx}(k, d) = E_{tx} - elec(k) + E_{tx} - amp(k, d)$$
$$= E_{elec} \times k + \varepsilon amp \times k \times d \qquad (15)$$

The radio energy consumed for receiving a k-bit message is determined by:

$$E_{RX}(k) = E_{RX} - elec(k)$$
$$E_{RX}(k) = E_{Elec} \times k \qquad (16)$$

As a result, the amount of overall energy used to transmit a k-bit message can be written as:

$$E_{ij} = E_{tx}(k, d) + E_{RX}(k) \qquad (17)$$

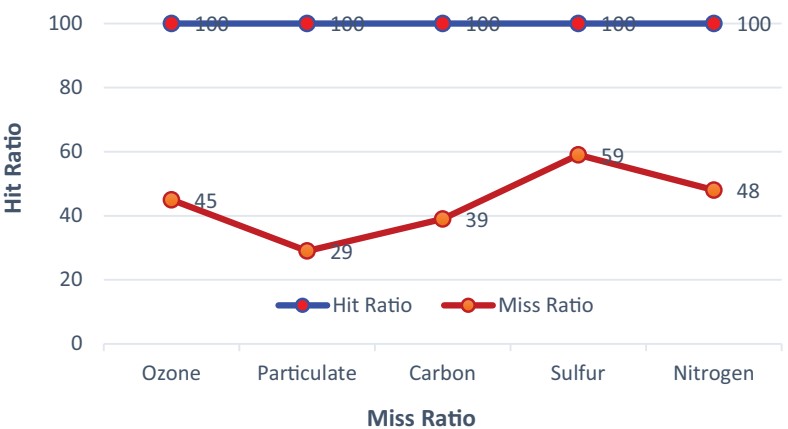

**Figure 4** **The comparison of static and dynamic edge nodes.**

A 64-bit operating system, Windows 10, equipped with an Intel Core i5-4300 CPU running at 1.90 GHz (4CPUs) and 8 GB of RAM, was utilized for this simulation. The simulation was conducted in JavaScript (JS). For coverage, 200 sensor nodes were distributed in a 500 m × 500 m area, with 10 battery-powered fog nodes placed within a radius of 50 to 300 m. Each fog node supports five services. The data sets (Weather Data) used were original and taken from http://iot.ee.surrey.ac.uk/, specifically from Brasov, Romania, covering the data collection period of 02/2014 to 06/2014, as mentioned on the website.

## RESULTS AND DISCUSSIONS

The proposed model is trained in a fully supervised manner using advanced datasets, each consisting of ten fog nodes and two hundred sensor nodes. The nodes are configured to predict and activate services based on pre-defined threshold values specific to different data sets. The model represents active nodes with predicted service-wise graphs showing both hit and miss ratios. The fog nodes highlight optimal services where the prediction is hazardous, and the required services are activated based on priority threshold values. The graphical illustration of static and dynamic edge nodes before reconfiguration is shown in Fig. 4.

In Fig. 5, fog node 1–10 demonstrates that most of the services are activated, achieving an accuracy of almost 92 percent in the prediction of services. In Fig. 5.1, all services are activated, achieving an impressive hit ratio of around 99%. Similarly, in Fig. 5.2, all services are activated except for particulate matter, resulting in the hit ratio approaching its minimum value. Overall, each figure in Fig. 5 illustrates the hit-and-miss ratios for the considered services, with varying hit ratios depending on the services' activation status. All the services are activated based on the demand for the required services. The percentage of hit-and-miss ratios varies depending on the sensor environment.

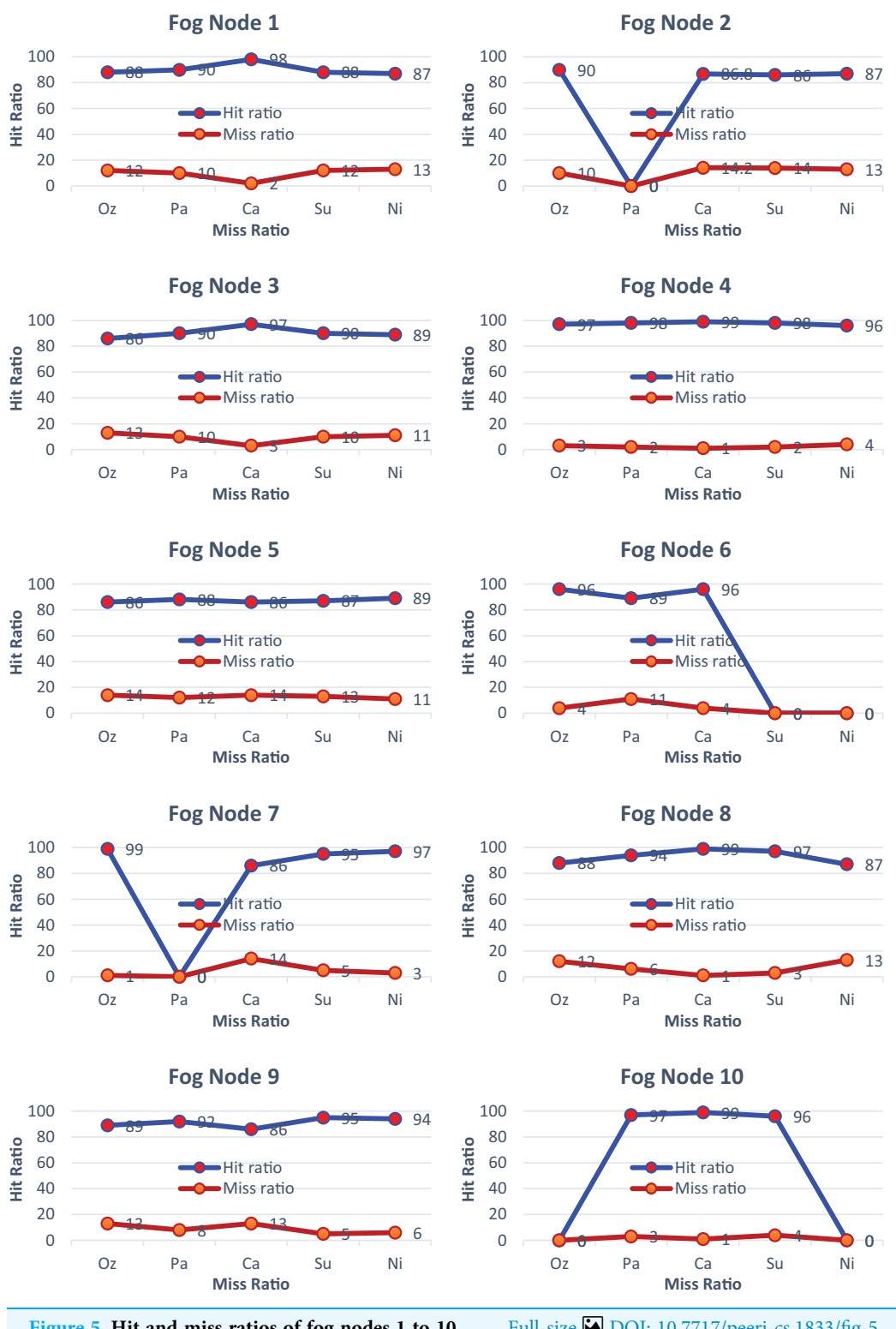

**Figure 5 Hit and miss ratios of fog nodes 1 to 10.**

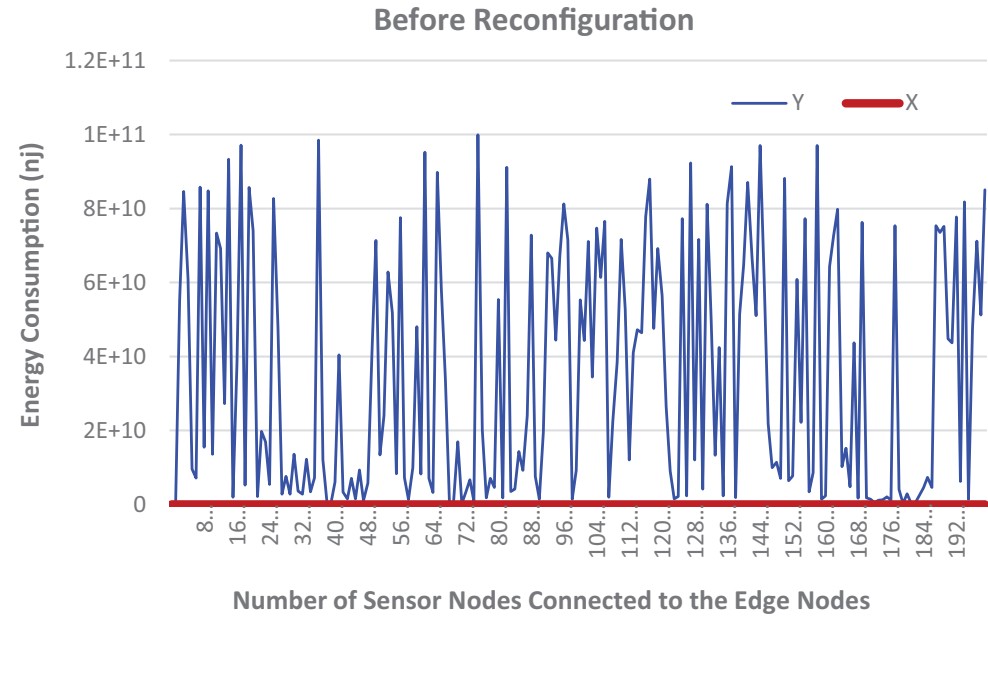

(a)

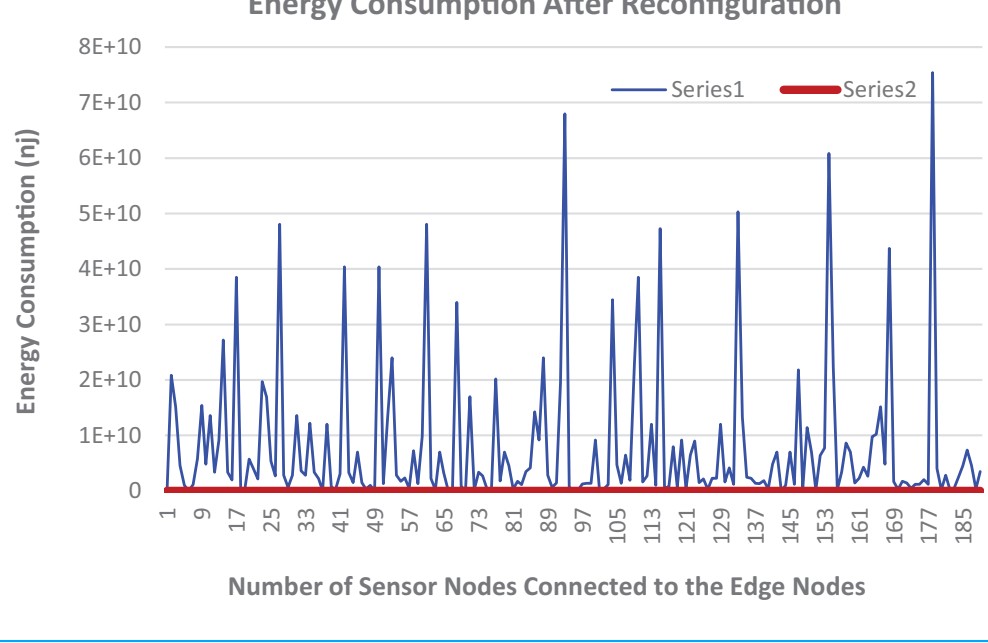

(b)

**Figure 6** Energy consumption (A) before and (B) after dynamic reconfiguration.

Figure 6A shows the graphical illustration of a static fog node with fixed services. It can be observed that the energy consumption for such a configuration is very high. Additionally, the energy consumption for each sensor is higher when compared to the dynamic fog node reconfiguration shown in Fig. 6B. An IIoT consists of a diverse number and types of devices with different functionalities and requires various types of services to evaluate the sensed data. Owing to their inherent limitations in terms of computational

power and memory capacity compared to cloud-based resources, edge nodes face challenges in simultaneously accommodating the complete suite of necessary services. To address this issue, efficient reconfiguration of edge nodes is employed to increase the service hosting capacity of a particular edge node at a specific time slot. We have evaluated the energy consumption of both fog nodes with fixed services and fog nodes with reconfiguration. It is evident that the energy consumption using the proposed DT-ID3 algorithm is significantly reduced compared to the energy consumption of static nodes with fixed services.

## CONCLUSIONS

The smart city network is made up of a variety of devices with varied functions that require different resources to evaluate the collected data. In comparison to cloud computing, the fog nodes have limited computational power and memory capacity. Hosting all necessary resources simultaneously in fog computing is challenging, and efficient fog node reconfiguration can improve the overall performance. Fog node reconfiguration can reduce energy usage and packet loss while increasing the hit ratio of required services. Efficient fog node reconfiguration is an interesting and demanding task for IoT-based smart cities, industries, and healthcare. A DT-based reconfigurable fog node system can allocate requested services based on availability or use DT methods to reconfigure the fog node according to the sensor environment. The proposed model can reduce energy consumption by over 50% and achieve a notable increase in the hit ratios of fog nodes, scaling up to 99% for the majority of services. Concurrently, there is a substantial reduction in miss-ratios, and the model efficiently adapts to dynamic or evolving environments. The synchronization of fog nodes and improving the hit ratio of services on demand are potential areas for future research. The proposed solution could handle both simple and composite services, and future work will focus on developing an optimal halting method for manipulating composite services.

### Funding

The work was supported by the financial supports from National Key Research and Development Program of China (2020YFB1805801), the National Natural Science Foundation of China (62171227, 62225503, 61835005, 62205151, 62275127, U2001601, 61935005, U22B2010); Jiangsu Provincial Key Research and Development Program (BE2022079, BE2022055-2), the Natural Science Foundation of the Jiangsu Higher Education Institutions of China (22KJB510031), and the Startup Foundation for Introducing Talent of NUIST. The funders had no role in study design, data collection and analysis, decision to publish, or preparation of the manuscript.

### Grant Disclosures

The following grant information was disclosed by the authors:
National Key Research and Development Program of China: 2020YFB1805801.

National Natural Science Foundation of China: 62171227, 62225503, 61835005, 62205151, 62275127, U2001601, 61935005, U22B2010.
Jiangsu Provincial Key Research and Development Program: BE2022079, BE2022055-2.
The Natural Science Foundation of the Jiangsu Higher Education Institutions of China: 22KJB510031.
The Startup Foundation for Introducing Talent of NUIST.

## Competing Interests

The authors declare that they have no competing interests.

## Author Contributions

- Rahat Ullah conceived and designed the experiments, analyzed the data, prepared figures and/or tables, authored or reviewed drafts of the article, and approved the final draft.
- Muhammad Yahya conceived and designed the experiments, performed the experiments, analyzed the data, performed the computation work, prepared figures and/or tables, authored or reviewed drafts of the article, and approved the final draft.
- Leonardo Mostarda analyzed the data, authored or reviewed drafts of the article, and approved the final draft.
- Abdullah Alshammari performed the experiments, performed the computation work, authored or reviewed drafts of the article, and approved the final draft.
- Ahmed I Alutaibi performed the experiments, analyzed the data, authored or reviewed drafts of the article, and approved the final draft.
- Nadeem Sarwar performed the experiments, analyzed the data, performed the computation work, authored or reviewed drafts of the article, and approved the final draft.
- Farhan Ullah analyzed the data, authored or reviewed drafts of the article, and approved the final draft.
- Sibghat Ullah conceived and designed the experiments, performed the experiments, analyzed the data, performed the computation work, prepared figures and/or tables, authored or reviewed drafts of the article, and approved the final draft.

## Data Availability

The complete dataset is available at Citypulse: http://iot.ee.surrey.ac.uk:8080/datasets.html#pollution.

They are available in raw format, or are semantically annotated using the citypulse information model.

Further details include the duration: 8/2014 - 10/2014, 2/2014 - 6/2014, 8/2014 - 9/2014, and the locations: Aarhus, Denmark (Open Data Aarhus), Brasov, Romania, Brasov, Romania, and the type of data is "Generated".

## Supplemental Information

Supplemental information for this article can be found online at http://dx.doi.org/10.7717/peerj-cs.1833#supplemental-information.

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
