# Peer review of "Intelligent decision making for energy efficient fog nodes selection and smart switching in the IOT: a machine learning approach"

_PeerJ Computer Science, doi:10.7717/peerj-cs.1833_

## Round 0.1 · original submission · Major Revisions

The authors should revise the article and address issues in the sections of research methods and modeling.

**Language Note:** The review process has identified that the English language must be improved. PeerJ can provide language editing services - please contact us at copyediting@peerj.com for pricing (be sure to provide your manuscript number and title). Alternatively, you should make your own arrangements to improve the language quality and provide details in your response letter. – PeerJ Staff

Reviewer 1 ·

Basic reporting

Please comments in additional comments section.

Experimental design

Please comments in additional comments section.

Validity of the findings

Please comments in additional comments section.

Additional comments

Thank you for the opportunity to review this article. Please find my comments listed below:
1. In Figure 4.1 and 4.10, the values of the x-axis and y-axis appear identical, with only a change in node names. This graphical representation can be somewhat confusing, and further clarification is needed. It is essential to clarify whether the configurations of these nodes differ or if these sensor nodes are placed within varying environmental contexts. Alternatively, it would be valuable to understand if these figures are depicting connections to fog nodes, which might clarify the apparent uniformity.
2. I am intrigued by the decision-making process of fog nodes in adjusting the sensor node configurations. However, the parameters governing this decision-making process remain somewhat unclear. Providing insights into these parameters would offer a deeper understanding of the system's inner workings.
3. While the paper mentions the use of the Decision Tree algorithm, it would be beneficial to delve deeper into the rationale behind this choice. A more detailed explanation of why a Decision Tree was chosen over alternatives, such as a Deep Learning approach, would be beneficial in justifying the selection.
4. Additionally, It is crucial to provide an in-depth explanation of the dataset used for training the Decision Tree Algorithm. A comprehensive description of the dataset, its sources, size, and any preprocessing involved would greatly enhance the paper's clarity.
5. In the context of the dataset, it is important to clarify whether there are multiple datasets specific to different services or if a single, all-encompassing dataset is employed across all services. This distinction can impact the generalizability and applicability of the proposed approach.
6. Overall, the paper introduces interesting concepts, but it would be valuable to include a brief summary at the end of each section, outlining the key points. This can assist readers in retaining the main points as they navigate through the paper.

Reviewer 2 ·

Basic reporting

The author prepared a comprehensive article. However, i have some major concerns regarding proposed model.
First , the author used decision tree to forecast the occurence of requests. I am wonder, how authors modeled it with machine learning? Did author used any well known technique or build their specific model. If specific, the article needs more details. Moreover, the abstract lacks in the description of actual contribution.
Second, authors improved hit ratio. What it means? Is it mean cache hit ? or something else? it seems to be too confusing.
Third, Proposed methodology needs improvement. Symbols are not well defined.
Fourth, the sections are not well coherent.
Fifth, what is the necassity of section 3? As, these are well known terms and equations. Authors need to reformualte with their own constraints.
The technical writing needs a lot of improvement.
The introduction should describe the structure of the article in last para.
Section 2 and 3 should be integrated and written as system model.

Experimental design

Needs improvement, as authors did not describe the modeling of decision tree as well as specific machine learning model.

Validity of the findings

The results seems to be valid.

Additional comments

Nil

Reviewer 3 ·

Basic reporting

1. Evaluation measures of the approach can be discussed in two or three lines at the end of the abstract section which will provide prospective readers with a better understanding of the research's focus and expected outcomes.
2. While the paper discusses the use of the Decision Tree Algorithm, it would be valuable to complement this with a comparative analysis against other machine learning models, such as Random Forest. A side-by-side evaluation can clarify the strengths and weaknesses of the selected approach, aiding in a more comprehensive assessment of its performance.
3. Have you explored the possibility of incorporating a time assessment in your research? I would suggest to introduce a time assessment as a component of the decision-making process for edge node selection. Subsequently, the results obtained through this assessment could be compared with cloud-based alternatives.
4. It might be beneficial to provide a brief explanation of how the paper's findings and approach contribute to the broader field of IoT and edge computing.

Experimental design

N/A

Validity of the findings

N/A

---

## Round 0.2 · accepted · Accept

The reviewers are satisfied with the revisions of the article. It is accepted in current form.

Reviewer 2 ·

Basic reporting

The author prepared the manuscript according to the comments. I have no further comments.

Experimental design

The author prepared the manuscript according to the comments. I have no further comments.

Validity of the findings

The author prepared the manuscript according to the comments. I have no further comments.

Reviewer 3 ·

Basic reporting

The authors have addressed all the comments and now the article is ready for publication in its current form.

Experimental design

The authors have addressed all the comments and now the article is ready for publication in its current form.

Validity of the findings

The authors have addressed all the comments and now the article is ready for publication in its current form.

Additional comments

The authors have addressed all the comments and now the article is ready for publication in its current form.